# Comprehensive Study on Environmental Behaviour and Degradation by Photolytic/Photocatalytic Oxidation Processes of Pharmaceutical Memantine



**Sandra Babić** [1,*] , **Davor Ljubas** [2] , **Dragana Mutavdžić Pavlović** [1], **Martina Biošić** [1], **Lidija Ćurković** [2,*] and **Dario Dabić** [1,3]

1 Faculty of Chemical Engineering and Technology, University of Zagreb, Trg Marka Marulića 19, 10000 Zagreb, Croatia
2 Faculty of Mechanical Engineering and Naval Architecture, University of Zagreb, Ivana Lučića 1, 10000 Zagreb, Croatia
3 Croatian Meteorological and Hydrological Service, Ravnice 48, 10000 Zagreb, Croatia
* Correspondence: sandra.babic@fkit.unizg.hr (S.B.); lidija.curkovic@fsb.hr (L.Ć.)

**Abstract:** Memantine is a pharmaceutical used to treat memory loss, one of the main symptoms of dementia and Alzheimer's disease. The use of memantine is expected to continue to grow due to the increasing proportion of the elderly population worldwide. The aim of this work was to conduct a comprehensive study on the behaviour of memantine in the environment and the possibilities of its removal from wastewater. Abiotic elimination processes (hydrolysis, photolysis and sorption) of memantine in the environment were investigated. Results showed that memantine is stable in the environment and easily leached from river sediment. Therefore, further investigation was focused on memantine removal by advanced oxidation processes that would prevent its release into the environment. For photolytic and photocatalytic degradation of memantine, ultraviolet (UV) lamps with the predominant radiation wavelengths of 365 nm (UV-A) and 254/185 nm (UV-C) were used as a source of light. TiO₂ in the form of a nanostructured film deposited on the borosilicate glass wall of the reactor was used for photocatalytic experiments. Photodegradation of memantine followed pseudo-first-order kinetics. The half-life of photocatalytic degradation by UV-A light was much higher (46.3 min) than the half-life obtained by UV-C light (3.9 min). Processes degradation efficiencies and evaluation of kinetic constants were based on the results of HPLC-MS/MS analyses, which also enable the identification of memantine oxidation products. The acute toxicity of the reaction mixture during the oxidation was evaluated by monitoring the inhibition of the luminescence of *Vibrio fischeri* bacteria. The results showed that memantine and its oxidation products were not harmful to *Vibrio fischeri*.

**Keywords:** memantine; hydrolysis; photolysis; sorption; photocatalysis; sol-gel TiO₂ film; degradation products; toxicity

## 1. Introduction

With the increase in the aging population worldwide, Alzheimer's disease and other dementias have become a rapidly increasing public health concern, with an estimated 50 million people currently living with dementia [1]. The prevalence of Alzheimer's disease is approximately 0.6% at the age of 60 but it doubles every five years, so the prevalence is about 40% at the age of 90 [2]. Currently, no cure exists for Alzheimer's disease but there are treatments that temporarily slow down the development of symptoms and improve cognitive functions. In Europe and the USA two symptomatic treatments are approved, the use of acetylcholinesterase (AChE) inhibitors and *N*-methyl-*D*-aspartate receptor antagonist memantine (3,5-dimethyladamantan-1-amine) [1]. Once in the organism, memantine is

poorly metabolized and most (57–82%) of the administered dose is excreted unchanged in urine [3]. Memantine is a compound highly soluble in water (29.4 mg/L, [4]).

In view of this, it is reasonable to expect that memantine will end up in wastewater and, without proper treatment, in environmental waters. Memantine was detected in rivers and sewage treatment plant (STP) influents and effluents in Japan at high frequency (>70%), with a maximal measured environmental concentration in seven rivers of 47.4 ng/L. The measured concentration of memantine in STP was lower than 1 μg/L with an average removal rate in three STPs lower than 20% [5]. Memantine was detected in effluent wastewater from a wastewater treatment plant (WWTP) located near Barcelona (Spain) at a concentration ranging from 0.028 to 0.134 μg/L for samples taken on 10 different days [6]. Memantine was also detected in sewage effluents in three Sweden STPs in concentrations ranging from 10 to 14 ng/L. It is also detected in the plasma of fishes exposed to sewage effluents (from <LOQ (0.5 ng/mL) to 2.3 ng/mL) [7]. Kårelid et al. [8] studied the adsorption of pharmaceuticals on granular activated carbon (GAC) and powdered activated carbon (PAC) at three Swedish wastewater treatment plants and observed that, among 22 investigated pharmaceuticals, only memantine showed removal lower than 95%. Despite the evidence that memantine is present in the environment, data on the fate and behaviour of memantine in the environmental are scarce.

The incomplete removal of pharmaceuticals in conventional wastewater treatment plants clearly indicates the need for the development of innovative technologies such as advanced oxidation processes (AOPs). AOPs have been proposed as a tertiary treatment for wastewater [9,10]. Among different AOPs, heterogeneous photocatalysis is a promising method for removing organic micropollutants (OMPs), including pharmaceuticals [11–13]. The most commonly used semiconductor photocatalyst is $TiO_2$ with the potential for the total mineralization of OMPs, resulting in the formation of non-toxic compounds ($CO_2$, $H_2O$ and the corresponding mineral acids). $TiO_2$ can be used in the form of $TiO_2$ powder suspension (slurry) or it can be immobilized by different techniques on different substrates such as borosilicate glass [14,15], alumina foam [16], alumina ring and borosilicate ring. Immobilization of $TiO_2$ on the adequate reactor walls eliminates the need to separate the photocatalyst from the treated water. Among different deposition techniques (sol-gel, thermal treatment, pulsed laser deposition, reactive evaporation, physical vapour deposition (PVD), chemical vapour deposition (CVD), electrodeposition, sol-spray, hydrothermal deposition, etc.), the sol-gel technique offers many advantages: relatively low cost, low processing temperature, simple deposition, relatively simple control of composition, possibility of various forming processes, and ability to prepare nano-sized thin films and to produce fine structures [14,17]. Sol-gel films can be generally deposited by two methods—the dip coating and the spin coating technique [18].

The aim of this study was to investigate the environmental behaviour of memantine and possibility of its degradation by advanced oxidation processes. Environmental behaviour was investigated by studying abiotic elimination processes: hydrolysis, photolysis and sorption. For investigation of photolytic/photocatalytic oxidation of memantine, a photoreactor with UV-A and UV-C lamps, and $TiO_2$ in a form of a nanostructured film deposited on borosilicate glass wall of the reactor was used. Process degradation efficiencies and evaluation of kinetic constants were based on the results of HPLC-MS/MS analyses, which also enable identification and monitoring of memantine degradation products. In addition, the acute toxicity of the reaction mixture during the degradation experiment was evaluated by monitoring the inhibition of the luminescence of *Vibrio fischeri* bacteria.

## 2. Results and Discussion

### 2.1. Environmental Behaviour

2.1.1. Hydrolytic and Photolytic Degradation

Hydrolytic degradation of memantine was investigated according to the procedure described in OECD 111 [19]. The results of hydrolytic degradation experiments showed that memantine is persistent to hydrolytic degradation with the degree of hydrolytic

degradation between 1.5% and 2.8% under the applied conditions (Figure 1A). Given that hydrolytic degradation of 10% at 50 °C corresponds to a half-life of approximately 30 days, which is equivalent to the half-life of 1 year at 25 °C [1], memantine was considered stable and no further investigation is required.

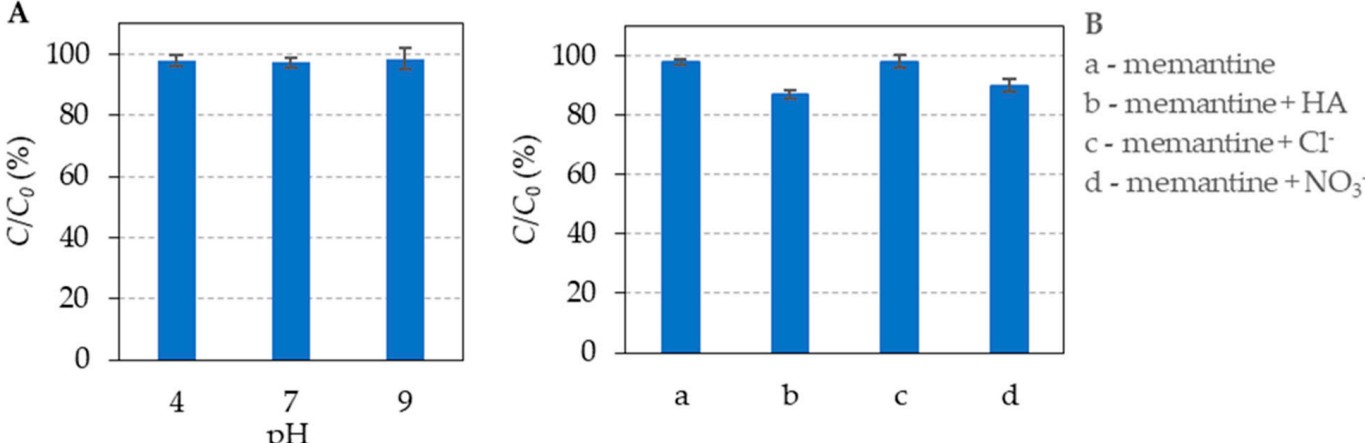

**Figure 1.** The degree of (**A**) hydrolytic degradation of memantine at 50 °C and (**B**) direct and indirect photolytic degradation (24 h exposure to artificial solar radiation, 10 mg/L memantine solution). Error bars refer to standard deviation.

Photolytic degradation was investigated with memantine solution in MilliQ water (direct photolysis) and in the presence of substances commonly present in environmental waters: humic acids (HA), chloride and nitrate (indirect photolysis). The concentrations of inorganic ions and humic acids were typical for the environment. The presence of HA and nitrates resulted in a lower concentration of memantine, while chlorides did not affect photolytic degradation of memantine. The photolysis due to the presence of HA and nitrate can be attributed to the formation of highly reactive hydroxyl radicals [20,21]. However, the observed decrease in the concentration of memantine was not significant (less than 15% after 24-h exposure to simulated solar radiation, Figure 1B), which points to the conclusion of its persistence during exposure to artificial solar radiation. Blum et al. [22] reported similar results of memantine photolytic persistence with a degree of degradation of less than 10%.

According to the available literature, similar environmental behaviour was not observed for other pharmaceuticals detected in environmental waters. They are usually susceptible to photolytic [20,21,23–25] or hydrolytic [26,27] degradation or to both elimination processes [28,29].

2.1.2. Sorption
Kinetics of Sorption and Desorption

Based on previously published papers [30,31], there is already some information about the tendency of memantine to sorption on soil and sediment particles, namely that sorption is definitely not a dominant process in its case. In this context, our goal was to compare from a kinetic point of view how much memantine is sorbed to the sediment particles and how much memantine is desorbed from the same sediment studied.

From Figure 2A, it can be seen that the "faster" sorption of memantine to the sediment sample occurs in the first 6 h, after which further sorption of memantine occurs slowly over the observed 24-h period. At the same time, desorption of the previously sorbed memantine takes place during the same time intervals (Figure 2B). It should be noted that, as the concentration of memantine increases, the amount of memantine sorbed and desorbed decreases, so that the largest difference between the amount sorbed and desorbed was obtained at the lowest initial concentration of memantine tested (0.1 mg/L). To investigate the control mechanisms of the sorption [32] and desorption processes, experiments were

performed at different time periods, i.e., a kinetic study was performed at three concentration levels (0.1, 0.5 and 2.0 mg/L), in contrast to the other six concentrations at which the sorption experiments will be carried out.

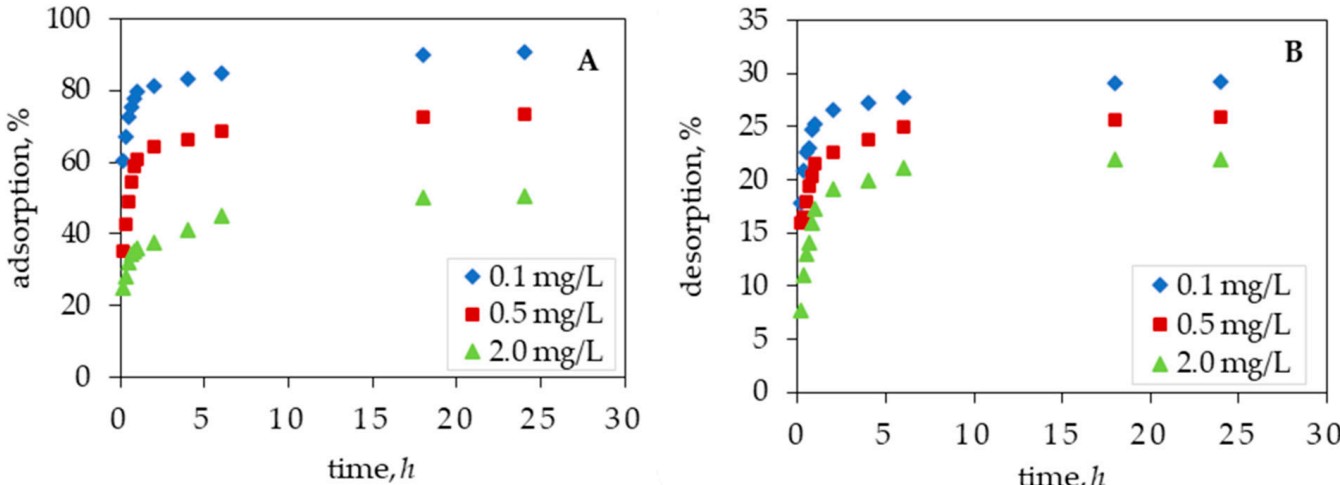

**Figure 2.** Kinetics of (**A**) sorption and (**B**) desorption for memantine on the sediment Studena, $T = 25\,^{\circ}\text{C}$.

The sorption and desorption data were analysed using three different kinetic models; Lagergren pseudo-first-order, pseudo-second-order and the intraparticle diffusion (IPD) model. All kinetic models are presented in Table 1 where $q_e$ and $q_t$ are the amounts of memantine ($\mu$g/g) adsorbed/desorbed on investigated sediment samples at equilibrium and at time $t$; $k_1$ (1/min) is the rate constant of the pseudo-first-order adsorption; $k_2$ (g/$\mu$g min) is the rate constant of the pseudo-second-order sorption and $k_{pi}$ ($\mu$g/g min$^{1/2}$) is the intraparticle diffusion rate parameter of stage $i$. $C_i$, the intercept of stage $i$, gives an indication of the thickness of the boundary layer, i.e., the larger the intercept, the larger the boundary layer effect.

**Table 1.** Kinetic models.

| Kinetic Model | Linear Form |
|---|---|
| Lagergren pseudo-first-order | $\ln(q_e - q_t) = \ln q_e - k_1 t$ |
| Ho's pseudo-second-order | $\frac{t}{q_t} = \frac{1}{k_2 q_e^2} + \frac{t}{q_e}$ |
| IPD model | $q_t = k_{pi}\sqrt{t} + C_i$ |

The sorption and desorption rate constants $k_1$, $k_2$ and $q_{e,\text{cal}}$ as well as the correlation coefficients ($R^2$) for the pseudo-first and pseudo-second models are shown in Table 2.

**Table 2.** Sorption and desorption kinetic parameters of memantine on the sediment Studena.

| Initial Concentration, mg/L | | $q_{e,exp}$, $\mu$g/g | Pseudo-First-Order | | | Pseudo-Second-Order | | |
|---|---|---|---|---|---|---|---|---|
| | | | $q_{e,calc}$, $\mu$g/g | $k_1$, 1/min | $R^2$ | $q_{e,calc}$, $\mu$g/g | $k_2$, g/$\mu$g min | $R^2$ |
| Sorption process | 2.0 | 10.11 | 13.33 | $2.303 \cdot 10^{-4}$ | 0.7666 | 10.24 | 0.0035 | 0.9990 |
| | 0.5 | 3.67 | 2.29 | $4.606 \cdot 10^{-4}$ | 0.5838 | 3.70 | 0.0162 | 0.9999 |
| | 0.1 | 0.91 | 0.25 | $9.212 \cdot 10^{-4}$ | 0.7508 | 0.91 | 0.0966 | 0.9998 |
| Desorption process | 2.0 | 4.40 | 3.74 | $-2.303 \cdot 10^{-4}$ | 0.8514 | 4.46 | 0.0107 | 1.000 |
| | 0.5 | 1.29 | 1.64 | $-2.303 \cdot 10^{-4}$ | 0.3369 | 1.30 | 0.0545 | 0.9999 |
| | 0.1 | 0.29 | 0.51 | $-1.612 \cdot 10^{-4}$ | 0.5115 | 0.29 | 0.3025 | 0.9999 |

From the results, it can be concluded that the pseudo-second-order kinetic model perfectly describes both the kinetics of memantine sorption and the kinetics of desorption of previously sorbed memantine, since very high correlation coefficients ($R^2 > 0.999$) were obtained in both cases. These results suggest that the sorption/desorption capacity is regulated by the number of available active sites on the sediment. According to this model, the maximum concentration absorbed at equilibrium ($q_e$) on the Studena sediment was approximately 10.11 µg/g, which corresponds to the maximum sorption capacity of this sediment for memantine in the experiments performed. However, if we consider the results of desorption according to the same model, it follows that the maximum concentration desorbed at equilibrium on the Studena sediment was about 4.40 µg/g, which practically corresponds to almost half of the amount of memantine previously sorbed. This ratio of sorbed/desorbed memantine from the sediment studied depends, of course, on the memantine concentration in contact with the sediment, so that Table 2 shows that, at concentrations of 0.1 mg/L and 0.5 mg/L, almost one-third is desorbed, which is less than the previously mentioned concentration of 2.0 mg/L, at which almost half was desorbed. The higher the concentration of memantine in contact with the sediment, the more memantine is washed out of the sediment, which is consistent with what was said before, i.e., that the sorption/desorption capacity is regulated by the number of available active sites on the sediment. The desorption rate constants according to the pseudo-second-order kinetic model are much higher than the sorption constants under the same conditions. Such a result is even more discouraging because, no matter how little memantine is sorbed on a sediment, it is still quite a lot and is rapidly desorbed, posing a risk of water contamination by memantine.

The kinetics of sorption and desorption can also be described from a mechanical point of view. The whole process of sorption and desorption can be controlled by one or more steps, such as surface diffusion, pore diffusion, external diffusion and sorption/desorption at the pore surface. When the sorbent is porous, as in the case of sediments, intraparticle diffusion often plays a major role. During rapid stirring, the diffusion mass transport can be related to the diffusion coefficient, which describes well the experimental sorption/desorption data. Results of the IPD model are shown in Table 3.

**Table 3.** Intraparticle diffusion model constants and correlation coefficients for memantine on the sediment Studena at different initial concentrations.

| Initial Concentration, mg/L | | Intraparticle Diffusion | | | | | | | | |
|---|---|---|---|---|---|---|---|---|---|---|
| | | **First Phase** | | | **Second Phase** | | | **Third Phase** | | |
| | | $k_{p1}$, µg/g min$^{1/2}$ | $C_1$ | $R^2$ | $k_{p2}$, µg/g min$^{1/2}$ | $C_2$ | $R^2$ | $k_{p3}$, µg/g min$^{1/2}$ | $C_3$ | $R^2$ |
| Sorption process | 2.0 | 0.5123 | 3.4451 | 0.9716 | 0.1853 | 5.4711 | 0.9819 | 0.0176 | 9.4375 | 1.000 |
| | 0.5 | 0.2895 | 0.8587 | 0.9934 | 0.0269 | 2.9168 | 0.9924 | 0.0062 | 3.4343 | 1.000 |
| | 0.1 | 0.0425 | 0.4792 | 0.9853 | 0.0046 | 0.7624 | 1.000 | 0.0014 | 0.8556 | 1.000 |
| Desorption process | 2.0 | 0.4058 | 0.3216 | 0.9915 | 0.0505 | 3.2505 | 0.9731 | 0.0009 | 4.3602 | 1.000 |
| | 0.5 | 0.0631 | 0.5692 | 0.9681 | 0.0142 | 0.9752 | 0.9975 | 0.0030 | 1.1806 | 1.000 |
| | 0.1 | 0.0156 | 0.1348 | 0.9673 | 0.0015 | 0.2490 | 0.9986 | 0.0003 | 0.2799 | 1.000 |

From these results, it is evident that the process of sorption and desorption of memantine on the studied sediment is multilinear, indicating that the sorption and desorption process occurs in three phases [33]: (i) initial boundary layer diffusion or adsorption/desorption at the outer surface, (ii) gradual intraparticle diffusion or diffusion in the pores, where the degree of intraparticle diffusion is rate-controlled, and (iii) equilibrium stage showing saturation of the sorbent surface.

This multilinearity of the sorption and desorption processes suggests that IPD was not the only rate-controlling step [34] but that multiple steps occur at this microlevel. All the ki-

netic results obtained are of great importance, especially the desorption information, which plays an important role in evaluating the behaviour of memantine in the environment.

Sorption Isotherms

The sorption of memantine was tested on three sediment samples (Pakra, Petrinjčica and Studena) and described by two sorption isotherms: the Linear and Freundlich sorption isotherms, and results are presented in Table 4. All presented values are expressed by the average value of three determinations. Achieved relative standard deviations are lower than 10%.

**Table 4.** Sorption coefficients ($K_d$), Freundlich and Dubinin-Radushkevich sorption isotherm parameters in 0.01 M $CaCl_2$ at initial pH values.

| Sediment Samples | Linear | | Freundlich | | | Dubinin-Radushkevich | | | |
|---|---|---|---|---|---|---|---|---|---|
| | $K_d$, mL/g | $R^2$ | $n$ | $K_F$, ($\mu$g/g)(mL/$\mu$g)$^{1/n}$ | $R^2$ | $\beta$, mol$^2$k/J$^2$ | $q_m$, $\mu$g/g | $E$, kJ/mol | $R^2$ |
| Pakra | 1.4267 | 0.9917 | 1.8776 | 1.6372 | 0.9189 | 0.0435 | 1.6394 | 3.39 | 0.7082 |
| Petrinjčica | 2.9658 | 0.9906 | 2.0467 | 3.1362 | 0.8894 | 0.0286 | 2.7194 | 4.18 | 0.6642 |
| Studena | 0.9771 | 0.9933 | 1.5868 | 1.0290 | 0.8932 | 0.0477 | 1.0377 | 3.24 | 0.6451 |

From the obtained regression coefficients $R^2$, it can be seen that only the linear isotherm describes the sorption process with $R^2 > 0.99$ in all cases, while the range of regression coefficients for the Freundlich isotherm is 0.89–0.92. The values of the Freundlich exponent, $n$, range from 1 to 10, indicating favourable sorption [35]. The Dubinin–Radushkevich model shows the worst agreement with the experimental data since $R^2$ ranges from 0.6451 to 0.7082. Since the values of sorption energy, $E$ (obtained from the D-R isotherm model), are from 3.24 to 4.18 kJ/mol for the investigated sediments, it could be said that the sorption of memantine on the investigated sediments is of a physical nature.

In addition to these three sorption models, many other sorption isotherm models were tried in the preliminary experiment to obtain more information about sorption of memantine, but without success. For example, when trying to applied the Langmuir isotherm we either obtained negative values for the Langmuir isotherm constants or the $R^2$ were extremely low ($R^2 < 0.25$). This indicates that the Langmuir sorption isotherm is not suitable to explain the sorption process of memantine on the sediment samples studied, since these Langmuir constants indicate the binding surface energy and monolayer coverage.

The values of the distribution coefficient $K_d$ from the linear isotherm and the adsorption capacity $K_F$ from the Freundlich isotherm indicate a slightly weaker binding of memantine to the sediments of Studena and Pakra compared to the sediment of Petrinjčica. However, it should be noted that high values of sorption coefficients were not expected at all, since, according to previous studies, a low tendency of memantine to sorption on soil and sediment particles was observed [30,31]. In any case, the results obtained in this study support the fact that memantine poses a major threat to natural waters because it is easily leached from sediment samples and thus has high mobility in these sediments.

Numerous previous studies clearly show that pH is one of the most important factors affecting the sorption mechanism and rate [36,37]. This is supported by the fact that whether the observed molecule behaves as a cation, anion or neutral molecule depends on the pH of the medium, but also that the activity/presence of metal ions present in the sediment changes depending on it. Since memantine as a molecule is characterized by a p$K_a$ constant (10.27), it can be concluded that it also occurs in ionic form. However, based on the natural pH value of the sediments studied, it could be concluded that memantine occurs exclusively as a neutral molecule in all sediments studied [30].

In addition to the effect of pH on the distribution of the ionic/molecular species of the memantine under environmental conditions, the effect of pH on their sorption at three different pH values (pH 5, 7 and 9) was also investigated (Figure 3A).

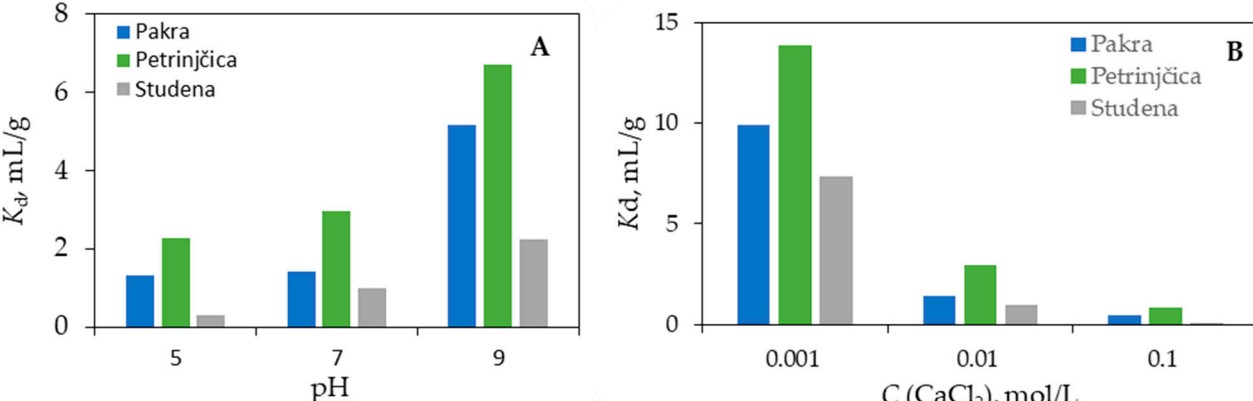

**Figure 3.** The influence of (**A**) pH and (**B**) ionic strength on the sorption capacity of memantine in studied sediments (T = 25 °C).

In these experiments, an inverse relationship between sorption and pH was observed for memantine [38,39], i.e., with higher acidity, the sorption coefficient decreases. In all three sediments studied, differences in $K_d$ values are observed with the change in pH. While the changes in distribution coefficient from neutral to the alkaline pH range are easily visible, the change in distribution coefficient from the neutral to the acidic pH range is much less pronounced. From these results, it can be clearly concluded that the influence of pH on the sorption of memantine dominates.

In addition to the influence of pH, the influence of ionic strength on the sorption of memantine was also investigated. It was found that the sorption coefficients decreased with increasing ionic strength (Figure 3B). The obtained results indicate a possible surface complexity between the memantine and the studied sediments. For all sediments studied, the highest $K_d$ coefficient values were obtained at the lowest concentration of CaCl$_2$ solution tested. The influence of ionic strength on sorption could be related to the fact that the thickness of the charged surface of the "electric double layer" is reduced, resulting in decrease in surface charge and fewer interactions between the ionic form of the drug and the sediment surface [40]. Of course, this theory is also supported by the fact that memantine is in the form of a neutral molecule [30] in all experiments performed, which makes a possible interaction even less likely. A similar trend was observed in a previous study of the sorption of memantine and in other studies of the sorption of pharmaceuticals [30,41–43].

*2.2. Photolytic and Photocatalytic Oxidation of Memantine in Aqueous Solution*

The photocatalytic activity of sol-gel TiO$_2$ film was evaluated through the degradation of memantine aqueous solution (10 mg/L) under ultraviolet (UV) lamps with the predominant radiation wavelengths of 365 nm (UV-A) and 254/185 nm (UV-C).

In order to investigate the kinetics of the photocatalytic degradation of memantine by photolytic and photocatalytic processes, the pseudo-first-order kinetic model was used. The linear form of the pseudo-first-order kinetic model is [44]:

$$ln\frac{C_0}{C_t} = -k_1 \cdot t \tag{1}$$

The half-life time $t_{1/2}$ was calculated using the following expression [16]:

$$t_{1/2} = \frac{ln\,(2)}{k} \tag{2}$$

where $C_t$ (mg/L) is the concentration of memantine at time $t$ (min), $C_0$ (mg/L) is the initial memantine concentration and $k_1$ (1/min) is the degradation rate constant.

The first-order degradation rate constant ($k_1$, 1/min) from equation 1 can be calculated by the slope of the straight line obtained from plotting linear regression of $-ln\,(C_t/C_0)$ versus irradiation time ($t$) (Figure 4A). Table 5 shows the pseudo-first-order kinetic constant ($k_1$, 1/min), coefficient of determination ($R^2$), half-life time ($t_{1/2}$, min) and efficiency ($\eta$, %)

for memantine removal by photolysis and photocatalysis. It is noticed that the pseudo-first-order model has an $R^2 > 0.96$, which confirms that the memantine removal follows a pseudo-first-order model.

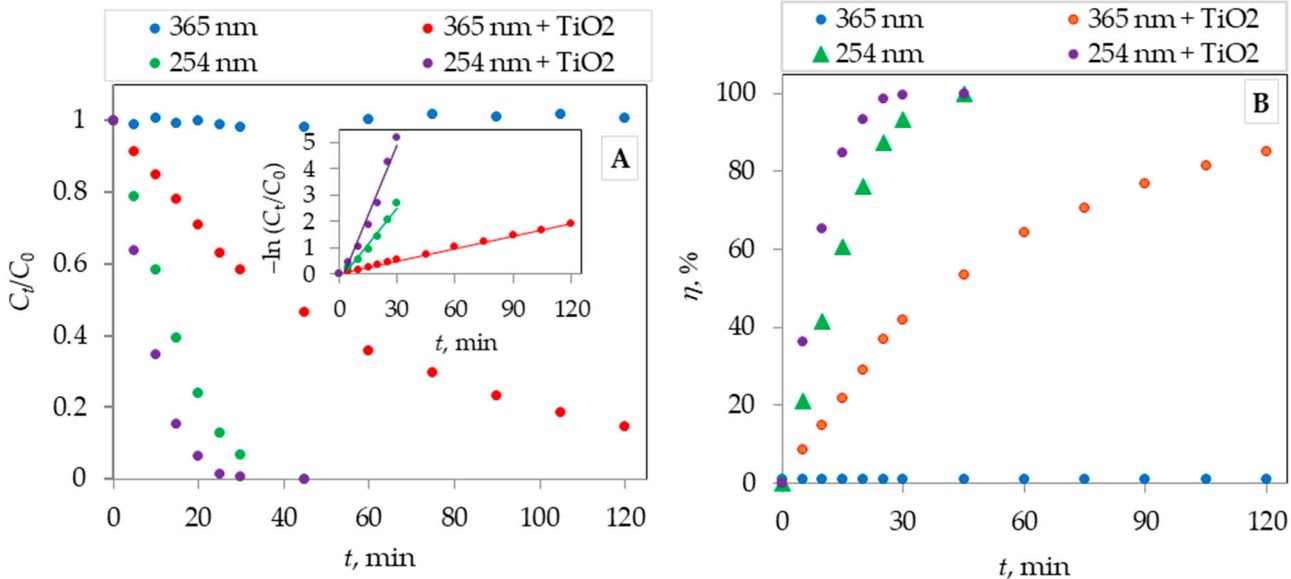

**Figure 4.** (**A**) Photolytic and photocatalytic degradation of memantine under UV-A (365 nm) and UV-C (254/185 nm) radiation by sol-gel nanostructured $TiO_2$ film as a function of irradiation time, $C_0$ (memantine) = 10 mg/L. Inset: linear transform of $-\ln (C_t/C_0)$ versus $t$. (**B**) Photolytic and photocatalytic degradation efficiency. All experiments were triplicated with the standard deviation from the average value ± 4%.

**Table 5.** Photolytic and photocatalytic degradation rate constants and half-lives of memantine.

| Experiment | $R^2$ | Regression Equation | $k_1$, 1/min | $t_{1/2}$, min | $\eta$, % |
|---|---|---|---|---|---|
| UV-A | – | – | – | – | 0 (after 120 min) |
| UV-C (254/185 nm) | 0.9686 | y = 0.0908x − 0.2293 | 0.0908 | 7.6 | 100 (after 45 min) |
| $TiO_2$ film + UV-C (254/185 nm) | 0.9693 | y = 0.1779x − 0.4418 | 0.1779 | 3.9 | 100 (after 45 min) |
| $TiO_2$ film + UV-A (365 nm) | 0.9984 | y = 0.0159x + 0.0277 | 0.0159 | 46.3 | 85 (after 120 min) |

From the photocatalytic experiments, it is observed that, under UV-A light (Figure 4A,B) after 45 min of irradiation, complete degradation of memantine is achieved (100% efficiency). Conversely, the photocatalytic experiments under UV-C light (Figure 4A,B) show that, after 120 min of irradiation, 85% of memantine removal is obtained. The photocatalytic degradation rate of memantine in the "UV-C + $TiO_2$ film" experiment is much faster (0.1779 min$^{-1}$) than that in the "UV-A + $TiO_2$ film" experiment (0.0159 min$^{-1}$). Similar behaviour was also observed in a recently published study of memantine oxidation [45]. In addition, photolytic oxidation of memantine by UV-A and UV-C light radiation was investigated. Memantine showed no degradation after 120 min of exposure to UV-A light alone, which is attributed to the low energy level of this type of UV light. Memantine degradation efficiency under exposure to UV-C light is the same as photocatalytic degradation by $TiO_2$ film irradiated with UV-C light (100% efficiency after 45 min, Table 5) but a rate constant of memantine degradation is two times lower for UV-C photolysis than for photocatalysis in the "$TiO_2$ film + UV-C" experiment (Table 5). The half-life of photolytic memantine degradation by UV-C light was almost two times higher (7.6 min) than the half-life obtained in the "$TiO_2$ film + UV-C" photocatalytic degradation experiment (3.9 min). It was found that the half-life of photocatalytic degradation by UV-A light was much higher (46.3 min) than the half-life obtained by UV-C light (3.9 min).

Švagelj et al. [16] published their findings on the photocatalytic degradation of memantine using a sol-gel TiO$_2$ film deposited on alumina foam substrate irradiated by UV-A light. The application of sol-gel TiO$_2$ film deposited on alumina foam substrate resulted in a larger specific surface area and therewith fast degradation of memantine can be obtained.

The diffuse reflectance spectroscopy (DRS) result and the Tauc plot are shown in Figure 5A,B. It was found that prepared TiO$_2$ only absorbs photons at wavelengths shorter than 400 nm. Based on the DRS, the Tauc plots can be obtained to determine the energy bandgap of TiO$_2$ (Figure 5B). It is observed that prepared TiO$_2$ presents a lower energy bandgap (2.99 eV) in comparison to commercial TiO$_2$ P25 (3.20 eV) [46].

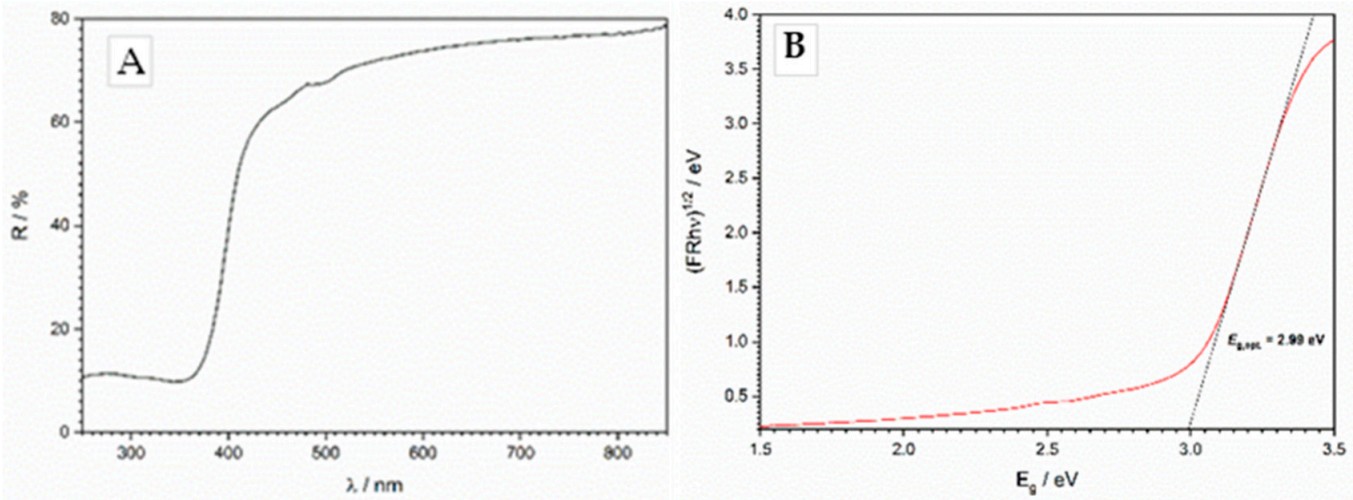

**Figure 5.** (**A**) DRS spectra and (**B**) Tauc plot for energy bandgap determination of TiO$_2$.

### 2.3. Oxidation Products of Memantine

Compared to the chromatogram of the memantine before oxidation, five new peaks were observed corresponding to the possible oxidation products of memantine. All five degradation products have lower retention times than memantine (Table 6), indicating that they are more polar. Tentative structures of memantine oxidation products (Table 6) were proposed based on the retention times, *m/z*-values and fragmentation patterns obtained through HPLC-MS/MS analysis. Mass spectra of memantine and its oxidation products are shown in Supplementary Materials, Figure S1.

Despite the different kinetics, all five degradation products were detected in all oxidation experiments, except photolysis UV-A light when degradation was not achieved. The same oxidation products after UV-C/H$_2$O$_2$ and UV-A/TiO$_2$ treatment were recently reported in [40].

### 2.4. Toxicity of the Mixture of Memantine and Its Degradation Products

Generally, some compounds do not show toxicity to a specific species, but this does not necessarily mean that they are not toxic or harmful to the environment or humans or another organism tested for toxicity [47,48]. Papac et. al. [45] determine that memantine was toxic to *Daphnia magna* (EC$_{50}$ = 7.19 mg/L). Blaschke et al. [49] demonstrate that chronic toxicity is not always more sensitive than acute toxicity. Keeping that in mind, assessment of the acute toxicity of memantine and its oxidation products during TiO$_2$ photocatalysis by UV-C light was carried out using *Vibrio fischeri* bacteria. Luminescence inhibition (%) was measured in triplicate for each tested sample and mean values and standard deviations (*s*) were calculated. The results presented in Table 7 indicate that memantine (10 mg/L) and its oxidation products were not harmful to *Vibrio fischeri* under the applied experimental conditions.

**Table 6.** Proposed chemical structures of memantine oxidation products.

| Compound | $t_R$, min | Chemical Formula | Chemical Structure |
|---|---|---|---|
| memantine [M+H]$^+$ $m/z$ 163 $m/z$ 107 | 14.7 | $C_{12}H_{22}N$ $C_{12}H_{19}$ |  |
| DP-1 [M+H]$^+$ $m/z$ 179 $m/z$ 135 | 2.87 | $C_{12}H_{22}NO$ $C_{12}H_{19}O$ $C_{10}H_{15}$ |  |
| DP-2 [M+H]$^+$ $m/z$ 177 $m/z$ 149 | 4.77 | $C_{12}H_{20}NO$ $C_{12}H_{17}O$ $C_{10}H_{13}O$ |  |
| DP-3 [M+H]$^+$ $m/z$ 193 $m/z$ 164 $m/z$ 135 | 2.29 | $C_{12}H_{20}NO_2$ $C_{12}H_{17}O_2$ $C_{11}H_{18}N$ $C_{10}H_{15}$ |  |
| DP-4 [M+H]$^+$ $m/z$ 195 $m/z$ 179 $m/z$ 135 | 4.11 | $C_{12}H_{22}NO_2$ $C_{12}H_{19}O_2$ $C_{12}H_{19}O$ $C_{10}H_{15}$ |  |
| DP-5 [M+H]$^+$ $m/z$ 209 $m/z$ 193 $m/z$ 135 | 1.53 | $C_{12}H_{20}NO_3$ $C_{12}H_{17}O_3$ $C_{12}H_{17}O_2$ $C_{10}H_{15}$ |  |

**Table 7.** Luminescence inhibition during the $TiO_2$ photocatalysis by UV-C light.

| Exposure Time, min | 0 | 10 | 20 | 30 | 45 | 60 | 180 |
|---|---|---|---|---|---|---|---|
| Luminescence inhibition $\pm s$, % | $0.86 \pm 0.02$ | $1.04 \pm 0.03$ | $1.40 \pm 0.05$ | $0.76 \pm 0.03$ | $0.67 \pm 0.04$ | $0.73 \pm 0.04$ | $1.98 \pm 0.07$ |

## 3. Environmental Relevance

The fate and behaviour of pharmaceuticals in the environment is controlled by their physicochemical properties and the characteristics of the environment. Once in the environment, the pharmaceutical can be distributed between different compartments of the environment (such as water, soil, air and biota) and be exposed to different biotic and abiotic elimination processes that can potentially lead to lowering of their environmental concentration. On the other hand, the results of elimination processes can lead to the formation of new compounds—degradation products—with different physicochemical and toxic properties.

Memantine is highly soluble in water (Table S1). From the results of in silico prediction of memantine biodegradability using different QSAR models provided by EPISuit [50], memantine can be considered a persistent compound because it does not biodegrade fast (biowin 2 < 0.5 or biowin 6 < 0.5) and its ultimate biodegradation timeframe prediction is longer than months (biowin 3 < 2.25) [51] (Table S1).

The results of our research showed that memantine is persistent to hydrolytic degradation. Although some photolytic degradation was observed in the presence of humic acids and nitrates, the degree of photodegradation was insignificant. Due to the absence of chromophores in the molecule of memantine, such results are expected. For persistent compounds, such as memantine, it is important to investigate their potential mobility in the environment determined by the compound's water solubility and sorption properties. Results of sorption/desorption experiments showed that memantine has a low tendency to sorption and is easily leached from river sediments. Considering this and its high solubility in water, it is possible to conclude that memantine will not be eliminated by natural processes and has the potential to be transported from the release site.

Such compounds, persistent and mobile in the environment, are of great concern for water quality since they are highly polar and are not removed from water by sorption. They can therefore end up in drinking water, posing a potential risk to human health [52]. Conventional wastewater treatment, based on microbial degradation and sorption, is expected to be ineffective for the removal of persistent and mobile compounds, since they are neither biodegradable nor sorbed substantially [52]. Therefore, it is of great importance to prevent their release into the environment by developing advanced and effective wastewater treatment processes.

Today, water treatment technology is trying to introduce processes that will be efficient and cheap, but also in accordance with ecological principles. One of the methods for degradation of such persistent and mobile compounds in water that is close to meeting these requirements is photocatalytic oxidation, with the use of titanium(IV) oxide ($TiO_2$) as a photocatalyst [53]. In addition to $TiO_2$, the presence of a suitable source of UV radiation that starts the process and oxygen dissolved in water are also necessary. It is a process that is included in the so-called advanced oxidation processes (AOPs). For the photocatalytic oxidation process, it is not necessary to add any additional chemicals except a solid photocatalyst (in the form of particles or nanostructured films on the reactor walls) and oxygen, while ensuring irradiance in UV spectra [54]. The use of solar radiation as a process activator (i.e., a source of UV radiation) and oxygen from the air around the reactor contribute to approaching the ecological principles of this technology. Photocatalytic oxidation of memantine resulted with the occurrence of five degradation products. According to the shorter chromatographic retention times compared to memantine, it is assumed that the oxidation products are more polar than memantine. This may indicate better solubility in water and a weaker tendency for sorption. Although oxidation products of memantine do not show inhibition of bioluminescence of *Vibrio Fischeri*, future studies on the current topic are suggested to assess the cytotoxicity and genotoxicity of memantine and its oxidation products in surface waters and wastewaters, as well as toxicological risks to ecosystems and human health. Furthermore, experimental data on the biodegradation of memantine and its oxidation products should be gathered.

## 4. Experimental Section

### 4.1. Materials and Chemicals

Analytical standard of memantine hydrochloride (CAS number: 41100-52-1) (Sigma Aldrich, St. Louis, MO, USA) of high purity (>98%) was used in this study. Memantine stock solution concentration of 1000 mg/L was prepared by weighing the accurate mass of memantine standard and dissolving it in methanol.

Acetonitrile, formic acid, citric acid, ascorbic acid and inorganic salts were of analytical grade and supplied by Kemika (Zagreb, Croatia). For buffer preparation, analytical grade reagents were used. Ultra-pure water was prepared using a Millipore Simplicity UV system (Millipore Corporation, Billerica, MA, USA).

For toxicity evaluation, freeze-dried and liquid luminescent bacteria *Vibrio fischeri* NRRL-B-11177 (LCK484, LUMINStox LUMISmini, Hach Lange, Varaždin, Croatia) were used. The bacterial reagents as well as reconstitution reagents were purchased from Kemika (Zagreb, Croatia).

For the preparation of $TiO_2$ sol (colloidal solution), the following components were used: titanium (IV) isopropoxide ($Ti(C_3H_5O_{12})_4$, TTIP, 97%, Sigma-Aldrich, St. Louis, MO, USA), i-propanol ($C_3H_7OH$, Grammol, Croatia), acetylacetone ($CH_3(CO)CH_2(CO)CH_3 \geq 99\%$, Honeywell, Charlotte, NC, USA), nitric acid ($HNO_3$, Carlo Erba Reagents, Barcelona, Spain) and polyethylene glycol ($H(OCH_2CH_2)nOH$, Mr = 5000–7000, Sigma-Aldrich, St. Louis, MO, USA). All these chemicals were analytical grade reagents.

### 4.2. Sediment Samples

Samples of river or fluvial sediments were collected on the territory of the Republic of Croatia and in the following areas: in Sisak-Moslavina County on the Petrinjčica River in the town of Petrinja, in Požega-Slavonia County on the Pakra River in the town of Pakrac and in Primorsko-Goranska County on the Studena River in the surroundings of the city Rijeka. In all locations, the samples were collected outside of human activities, which provides some assurance that the samples are not contaminated, especially in the case of pharmaceuticals, and in the summertime when it is easier to reach the area and collect samples due to dryness. All samples were air-dried, crushed, sieved through a 2-mm sieve and characterized according to the proposed procedure [55]. The physicochemical properties of used samples can be seen in the previously published work [31].

### 4.3. Hydrolytic Degradation Experiments

Hydrolytic degradation was conducted at (50 ± 0,1) °C (Incubator shakers KS 3000 i control, IKA, Staufen, Germany) for 5 days and at three pH values (4, 7 and 9) in capped glass vials under dark conditions. A buffer solution with a pH value of 4 was prepared by mixing 38.55 mL of 0.2 M $K_2HPO_4$ and 61.45 mL of 0.1 M citric acid. A pH value buffer solution of 7 was prepared by mixing 29.63 mL of 0.1 M NaOH, 50 mL of 0.1 M $KH_2PO_4$ and 20.37 mL of MilliQ water, and the pH 9 buffer solution was prepared by mixing 21.30 mL of 0.1 M NaOH, 50 mL of 0.1 M $H_3BO_3$ in 0.1 M KCl and 28.70 mL of MilliQ water. The pH of each buffer solution was checked with pH meter S20 SevenEasy (Mettler Toledo, Greifensee, Switzerland). Memantine solutions were prepared in appropriate buffer solutions at a concentration of 10 mg/L. Concentration of memantine solutions after hydrolytic degradation experiment were determined by HPLC-MS/MS. All experiments were performed in three replicates.

### 4.4. Photolytic Degradation Experiments at Environmentally Relevant Conditions

Direct photolysis experiment was performed with memantine solution in MilliQ water (10 mg/L). To test the possibility of indirect photolysis, solutions of memantine (10 mg/L) were prepared in solutions of $Cl^-$ ions (10 mg/L), $NO_3^-$ ions (3 mg/L) and humic acids (3 mg/L). Forty millilitres of test solution were irradiated in quartz vessels (diameter 4.6 cm) placed in Suntest CPS+ simulator (Atlas, Linsengericht, Germany). Suntest CPS+ is equipped with a temperature sensor and a xenon lamp as a source of artificial sunlight in the wavelength range of 300–800 nm. The distance between the liquid surface and the lamp was 14 cm. During the experiments, the radiation intensity was maintained at 500 W m$^{-2}$ and the reaction temperature was kept at (25 ± 2) °C. In all photolysis experiments, dark control experiments were performed under the same conditions but protected from the radiation. Control samples with the same composition as test solutions were used to establish that memantine degradation was a consequence of the irradiation. Aliquots of irradiated memantine solution were analysed by HPLC-MS/MS. All experiments were performed in three replicates.

### 4.5. Sorption Experiments

Batch sorption experiments were performed according to the OECD 106 procedure [56]. The procedure is performed in triplicate by shaking on a laboratory shaker (Innova 4080 Incubator Shaker, NewBrunswick Scientific, Edison, NJ, USA), which allows continuous contact of the sediment samples with the memantine solution. To avoid photolytic degra-

dation, all experiments were performed in the dark and, to avoid microbiological activities, all sediment samples were sterilized beforehand.

It is very important to choose a good ratio between sediment (sorbent) and memantine solution. Since previous studies [30,31] show that memantine does not have excessive sorption potential to sediment or soil samples, all experiments were performed with a 1:10 (*w/v*) sediment/memantine solution ratio. In a previously published paper [30], it was determined that 24 h was sufficient for memantine to reach sorption equilibrium, so all experiments were conducted with 24 h of shaking. The procedure consisted of adding 10 mL known concentration (0.1–2.0 mg/L) of memantine solution in 50 mL of laboratory glass to 1 g of air-dried sediment. The prepared suspension was shaken in a shaker (at 200 rpm) for 24 h at room temperature (25 °C), filtered through a 0.45 μm syringe filter and transferred to HPLC vials. Blank samples containing the same amount of sediment and soil in contact with 10.0 mL of 0.01 M CaCl$_2$ solution were also included in the analysis. They served as controls to detect interfering compounds or contaminated sediment.

In order to investigate the influence of pH and ionic strength on the sorption of memantine on the sediments studied, a series of experiments were performed in which one of the factors was changed while the others remained constant. The effect of pH was monitored using three series of experiments with different pH values of the studied memantine solutions (pH values 5, 7 and 9). All these experiments were performed in a 0.01 M CaCl$_2$ solution. To determine the effect of ionic strength, the pH of the memantine solution must be adjusted to the initial value (pH 7.0). These experiments were performed with three different concentrations of CaCl$_2$ solution (0.001, 0.01 and 0.1 mol/L).

Since the experiments to determine the required contact time (24 h) were the basis for the determination of sorption kinetics, we were able to start immediately with the determination of the kinetics of memantine desorption from the sediments studied. Desorption kinetics were studied for three memantine solutions in 0.01 mol/L CaCl$_2$ (0.1, 0.5 and 2.0 mg/L) using a decanting and refilling technique. After 24 h of shaking, the memantine solutions in contact with the sediment samples were replaced with fresh 0.01 mol/L CaCl$_2$. Memantine solution was removed using a disposable glass pipette. The residual solution that could not be removed before the desorption experiment was determined gravimetrically, and the same amount of 0.01 mol/L CaCl$_2$ as the removed memantine solution was weighed and added to the residual solution. Samples were then shaken at 25 °C for various periods (10, 20, 30, 40 and 50 min, and 1, 2, 4, 6, 12, 18 and 24 h).

### 4.6. Photolytic and Photocatalytic Oxidation Experiments

All experiments were carried out with 10 mg/L memantine solution in two borosilicate glass tubes (200 mm in height, 30 mm in diameter, 0.11 L) with continuous purging with air (O$_2$), at (25 ± 0.2) °C: (i) with the TiO$_2$ film, for photocatalytic experiments and (ii) without the TiO$_2$ film, for photolytic experiments. UV lamps (UV-A and UV-C) were placed in the middle of each reactor. Detailed experimental set-up is published elsewhere [17]. UV lamps used in experiments were 15 W mercury UV lamps: (i) model Pen-Ray CPQ-7427, UV-A with $\lambda_{max}$ = 365 nm and (ii) model Pen-Ray 90-0004-07, UV-C with $\lambda_{max}$ = 254/185 nm, manufactured by UVP (Upland, CA, USA). Both lamps were used with the same electrical source PS-4, *I* = 0.54 A, from UVP, too. The experimental set-up was described in detail in [17]. The reaction temperature was controlled by the circulation of cooling water. Total irradiation time for each oxidation test was 120 min. Aliquots of 1 mL were collected in defined time intervals and stored in the dark at 4 °C until HPLC-MS/MS analysis.

Nanostructured TiO$_2$ film was deposited on a borosilicate glass substrate by the sol-gel process using the dip-coating method. TiO$_2$ colloidal solution (sol) was prepared by mixing titanium(IV) isopropoxide (Ti-iPrOH) as a precursor, i-propyl alcohol (iPrOH) as a solvent, acetylacetone (AcAc) as a chelating agent, nitric acid (NA, 0.5 M) as a catalyst and polyethylene glycol as an organic/polymer additive in the amount of 2 g. The molar ratio of these reactants was Ti-iPrOH:iPrOH:AcAc:NA=1:35:0.63:0.015. The film was dried at 100 °C for 1 h prior to the deposition of the next layer. After the deposition of the three

layers, the deposited film was heat-treated at 550 °C for 4 h [14]. The procedure for the film preparation as well as its characterization was described in detail elsewhere [14,17].

Energy bandgap (Eg) of prepared $TiO_2$ was calculated from diffuse reflectance spectroscopy (DRS) measurements, which were performed on a QE Pro High-Performance Spectrometer (Ocean Insight, Orlando, FL, USA) equipped with an integrating sphere and a DH 2000 deuterium–halogen source in the analysis range 200–1000 nm with a resolu-tion of 1 nm and integration time of 10 s.

### 4.7. HPLC-MS/MS Analysis

Samples from photolytic and hydrolytic degradation experiments as well as samples from AOP experiments were analysed using an Agilent Series 1200 HPLC system (Santa Clara, CA, USA) coupled with an Agilent 6410 triple-quadrupole mass spectrometer equipped with an ESI interface (Santa Clara, CA, USA). Chromatographic separation was performed on an Kinetex C18 column (100 mm × 2.1 mm, 2.6 μm) (Phenomenex, Torrance, CA, USA) using mobile phase comprising MilliQ water with 0.1% formic acid as eluent A and acetonitrile with 0.1% formic acid as eluent B. The composition of 50% organic phase (B) was maintained at flow rate of 0.2 mL/min throughout the analysis. An injection volume of 5 μL was used in all analyses. The analyses were done in positive ion mode under the following conditions: drying gas temperature 350 °C; capillary voltage 4.0 kV; drying gas flow 11 L/min and nebulizer pressure 35 psi. Instrument control, data acquisition and evaluation were done with Agilent MassHunter 2003–2007 Data Acquisition for Triple Quad B.01.04 (B84) software (Santa Clara, CA, USA).

The residual concentration of memantine in the remaining liquid phase after sorption was analysed using UHPLC-MS (Agilent 6490 coupled with Agilent Infinity UHPLC System Triple Quadrupole Mass Spectrometer, Santa Clara, CA, USA) with electrospray ionization. The chromatographic column Shim pack XR ODS II (50 mm × 2 mm i.d., 1.6μm) (Shimadzu, Duisburg, Germany) was used at 30 °C with an injection volume of 1 μL. The mobile phase consisted of two eluents: eluent A (0.1% formic acid in MilliQ water) and B (0.1% formic acid in acetonitrile) and was performed in gradient elution mode. The gradient started with a 0.1-min linear gradient from 100% A to 10% B, followed by a 1.0-min linear gradient to 98% B, followed by a 0.5-min linear gradient back to 100% A held for 0.4 min. The flow rate was 0.2 mL/min. All analyses were performed in positive ion mode under the following parameters: drying gas temperature 200 °C, capillary voltage 3.0 kV, drying gas flow rate 15 L/min and nebulizer pressure 20 psi. Memantine was analysed by MRM, using the two highest characteristic precursor ion/product ion transitions ($m/z$ 291.25→230.2; $m/z$ 291.25→123.0).

### 4.8. Assessment of Acute Toxicity by Vibrio Fischeri

Acute toxicity assessment toward *Vibrio fischeri* culture was performed on standard solutions of memantine (10 mg/L), a mixture of memantine with its degradation products and finally the degradation products themselves without the detectable presence of memantine. Acute toxicity assessment was performed according to the method described in detail in [29]. In brief, sample solutions for toxicity measurements were prepared by serial dilutions in linear progression with addition of 2% NaCl. The experiments were conducted in a test tube by combining each volume of initial or diluted sample (1.5 mL) and 0.5 mL of bacterium *Vibrio Fischeri* suspension. The inhibition of luminescence was measured before and after 30 min of exposure of the sample to *Vibrio fischeri* on a luminometer (LUMIStox 300 Hach Lange, Düsseldorf, Germany) at 15 °C. In order to control bacteria performance, the reference substances $ZnSO_4$ x $(OH_2)_7$ (109.9 μg/mL) and $K_2Cr_2O_7$ (22.6 μg/mL) were used.

## 5. Conclusions

This research provides a comprehensive picture of memantine behaviour in the aquatic environment. The results showed that memantine is resistant to hydrolytic and photolytic (i.e., irradiated by solar light) degradation, has a low tendency to sorption and is easily desorbed from river sediments. For such compounds, which are persistent and mobile in the environment, it is of great importance to prevent their release into the environment by effective wastewater treatment.

Investigation of memantine oxidation by photolytic/photocatalytic oxidation by UV-A and UV-C light showed that memantine could be completely oxidized within 30 min during photocatalytic and within 50 min during photolytic oxidation processes by UV-C light. Photolytic degradation by UV-A light did not occur, while photocatalytic degradation by UV-A light did occur although the degradation rate was lower and memantine degradation was not completed even after 120 min. A kinetic study showed that the oxidation in all experiments, in which oxidation occurred, followed pseudo-first-order reaction kinetics.

As a result of the oxidation, five oxidation products were formed, which were identified using high performance liquid chromatography coupled with a triple quadrupole mass spectrometer. The same oxidation products were identified in all investigated processes in which oxidation occurred. The results of the acute toxicity assessment of memantine and its mixture with oxidation products indicate that the resulting products are not harmful.

**Supplementary Materials:** The following supporting information can be downloaded at: https://www.mdpi.com/article/10.3390/catal13030612/s1, Figure S1: Mass spectra of memantine and its oxidation products; Table S1: Results of EPISuite biodegradability prediction for memantine.

**Author Contributions:** Conceptualization, S.B. and D.L.; methodology, S.B., D.L., D.M.P., M.B., L.Ć. and D.D.; formal analysis, M.B., D.M.P. and D.D.; investigation, S.B., D.L., D.M.P., M.B., L.Ć. and D.D.; resources, S.B., D.L., D.M.P. and L.Ć.; data curation, S.B., D.L., D.M.P., M.B., L.Ć. and D.D.; writing—original draft preparation, S.B., L.Ć. and D.M.P.; writing—review and editing, S.B., D.L., D.M.P., M.B., L.Ć. and D.D.; visualization, L.Ć., D.M.P. and M.B.; supervision, S.B.; funding acquisition, S.B. All authors have read and agreed to the published version of the manuscript.

**Funding:** This research was funded by the Croatian Science Foundation under the project Fate of pharmaceuticals in the environment and during advanced wastewater treatment (PharmaFate) (IP-09-2014-2353).

**Data Availability Statement:** The data presented in this study are available upon reasonable request from the corresponding author.

**Conflicts of Interest:** The authors declare no conflict of interest.

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
