# Peer review of "Comprehensive Study on Environmental Behaviour and Degradation by Photolytic/Photocatalytic Oxidation Processes of Pharmaceutical Memantine"

_catalysts, doi:10.3390/catal13030612_

Round 1

Reviewer 1 Report

Comments

Manuscript Number: catalysts-2264736

Title: Comprehensive study on environmental behaviour and degradation by photolytic/photocatalytic oxidation processes of pharmaceutical memantine

This study investigated the behavior and removal of pharmaceutical memantine in the environment. The researchers found that memantine is persistent and mobile in the environment, with low sorption and leaching from river sediments. Photolytic and photocatalytic oxidation processes were effective in removing memantine from wastewater, with identified oxidation products being non-harmful

This work is interesting, which is a significant advancement over existing knowledge, but it needs substantial improvements before considering for publication. The publication is recommended and subjected to revision as mentioned below in comments to the authors:

Ø  Consider including a statement about the significance of the study's findings in the context of environmental health and safety.

Ø  Provide more information about the study’s limitations and suggest potential directions for future research.

Ø  Discuss the potential environmental implications of the study and how the findings could contribute to developing more efficient and sustainable wastewater treatment methods.

Ø  Acknowledging the experiment’s limitations, such as potential errors or uncertainties in the measurements, is essential. This can help readers interpret the results in the appropriate context.

Ø  Correct all spelling and grammatical mistakes

Ø  give proper referencing in the methodology section

Author Response

Response to Reviewer 1 Comments

This study investigated the behavior and removal of pharmaceutical memantine in the environment. The researchers found that memantine is persistent and mobile in the environment, with low sorption and leaching from river sediments. Photolytic and photocatalytic oxidation processes were effective in removing memantine from wastewater, with identified oxidation products being non-harmful.

This work is interesting, which is a significant advancement over existing knowledge, but it needs substantial improvements before considering for publication. The publication is recommended and subjected to revision as mentioned below in comments to the authors:

Point 1: Consider including a statement about the significance of the study's findings in the context of environmental health and safety. Provide more information about the study’s limitations and suggest potential directions for future research. Discuss the potential environmental implications of the study and how the findings could contribute to developing more efficient and sustainable wastewater treatment methods.

Response 1: The authors thank the Reviewer for this remark. A new section “Environmental relevance” was added with the discussion on the potential environmental implications of our study and the potential contribution of study findings to development of efficient wastewater treatment processes. Accordingly, new references were added.

Point 2: Acknowledging the experiment’s limitations, such as potential errors or uncertainties in the measurements, is essential. This can help readers interpret the results in the appropriate context.

Response 2: The authors thank the Reviewer for the observation. All the experiments were triplicated with the standard deviation from the average value ± 4%, therefore we did not add error bars.

Error bars were added only in Figure 1. and standard deviations were added in Table 7.

Point 3: Correct all spelling and grammatical mistakes.

 Response 3: Thank you very much for your observation. The manuscript was revised carefully and we corrected all spelling and grammatical mistakes.

Point 4: Give proper referencing in the methodology section.

Response 4: Thank you very much for your observation. The manuscript was checked carefully in the methodology section with the special focus to referencing.

Reviewer 2 Report

1, For the reader to obtain more experimental details, the HPLC-MS/MS data used for analyzing processes degradation efficiencies and evaluation of kinetic constants absorption spectrum must be included in the Supporting Information.

2. As the authors said all experiments were performed in three replicates. But there is no data statistical error bars in all Figures.  

Author Response

Response to Reviewer 2 Comments

Point 1: For the reader to obtain more experimental details, the HPLC-MS/MS data used for analyzing processes degradation efficiencies and evaluation of kinetic constants absorption spectrum must be included in the Supporting Information.

Response 1: The authors thank the Reviewer for the suggestion. Mass spectra of memantine and its oxidation products are added in Supplementary Materials, Figure S1.

Point 2: As the authors said all experiments were performed in three replicates. But there is no data statistical error bars in all Figures. 

 Response 2: The authors thank the Reviewer for the observation.

All the experiments were triplicated with the standard deviation from the average value ± 4%, therefore we did not add error bars.

Error bars were added only in Figure 1 and standard deviations were added in Table 7.

Reviewer 3 Report

I accept the manuscript to be published after minor revision. The research wok is interesting but some points can taken in consideration before publication

1-Titania supported on glass substrate can be investigated by physicochemical techniques as XRD, HRTEM to investigate the nanostructure and dimensions

2-What about surface area of photocatalyst

3-What about optical properties of the photocatalyst as DRS and PL measurement

4-Some sorption and kinetic equations as Elovich and Dubunin can adde to the revised manuscript to illustrate more clear mechanism on adsorption mechanism

5-The experimental work must be illustrated on more details

Author Response

Response to Reviewer 3 Comments

I accept the manuscript to be published after minor revision. The research wok is interesting but some points can taken in consideration before publication.

Point 1: Titania supported on glass substrate can be investigated by physicochemical techniques as XRD, HRTEM to investigate the nanostructure and dimensions.

Response 1: Thank the Reviewer for the comment. We have performed detail characterization of surface area of used TiO2 film by AFM, XRD, Micro-Raman, FTIR which we have published in previous manuscripts. We try to avoid repetition; therefore, we explain in section 3.6. Photolytic and photocatalytic oxidation experiments with our references: “The procedure for the film preparation as well as its characterization was described in detail elsewhere [14,17].”.

Point 2: What about surface area of photocatalyst.

Response 2: : Thank the Reviewer for the comment. We have performed detail characterization of surface area of used TiO2 film by AFM which we have published in previous manuscripts. We try to avoid repetition; therefore, we explain in section 3.6. Photolytic and photocatalytic oxidation experiments with our references: “The procedure for the film preparation as well as its characterization was described in detail elsewhere [14,17].”.

Point 3: What about optical properties of the photocatalyst as DRS and PL measurement

Response 3: The authors thank the Reviewer for the comment and given suggestions. We have performed DRS analysis, the diffuse reflectance spectroscopy (DRS) result and the Tauc plot are shown in Figure 5A and 5B. Device for photoluminescence spectroscopy (PL) analysis was not available to us, therefore we did not perform the PL analysis.

Point 4: Some sorption and kinetic equations as Elovich and Dubunin can adde to the revised manuscript to illustrate more clear mechanism on adsorption mechanism.

Response 4: Thanks to the reviewer for this comment. However, as I wrote earlier, other isotherm models besides the linear and Freundlich models have been tested. I mentioned the Langmuir model because it is one of the best known and very poor results were obtained for this model, but quite a number of other models were tested. Of the models you mentioned, I added the Dubinin-Radusckevich model because with this model the nature of sorption can be determined, so it makes sense to test it. However, the R2 of the model for the sediments studied is between 0.6 and 0.7, which means that it is not a reliable statement of physical sorption, although there is a high probability of it. Unfortunately, I did not consider Elovich's model, as it has the following R2 values: 0.03 (Pakra sediment), 0.65 (Petrinjčica sediment) and 0.22 (Studena sediment). All the mentioned values support the fact that this model cannot describe the sorption of memantine on the studied sediments.

Point 5: The experimental work must be illustrated on more details.

Response 5: The authors thank the Reviewer for the comment. The experimental work is escribed in more detail.

Round 2

Reviewer 2 Report

The authors have addressed my concerns. I recommend publishing this work.